# A Space-Time Absolute Nodal Coordinate Formulation Cable Element and the Study of Its Accuracy and Efficiency

Dekun Chen [1] , Kun Li [1], Nianli Lu [1,*] and Peng Lan [2,3]

1   School of Mechatronics Engineering, Harbin Institute of Technology, Harbin 150001, China;
    chendekun@hit.edu.cn (D.C.)
2   School of Mechanical & Electrical Engineering, Xi'an University of Architecture and Technology,
    Xi'an 710055, China
3   State Key Laboratory of Green Building in Western China, Xi'an University of Architecture and Technology,
    Xi'an 710055, China
*   Correspondence: n.lu@hit.edu.cn

**Abstract:** In this paper, a space-time absolute nodal coordinate formulation cable (SAC) element forming technique based on the Lagrange family of shape functions is proposed. Two distinct SAC elements, each with a distinct spatial shape function, have been generated by this method. Moreover, the external forces such as the bending moment and the air resistance formula have been accounted for. The Lagrange multiplier method, along with the concepts of replacement constraint and supplementary constraint, has been employed to provide a solution for the dynamics of constrained mechanical systems. Additionally, a constraint conversion strategy has been suggested. The solver has been constructed through Hamilton's law of varying action. The space-time finite element method is used to solve dynamic problems, employing the Newton algorithm and quasi-Newton algorithm. The accuracy and efficiency of the solution has been verified by three simulations and one experiment. The circle-bending static simulation and the double-ended velocity impact dynamic simulation demonstrate the accuracy of the two elements. The correlation between statics and dynamics has been studied for different discretization methods and different solvers' calculation accuracy and efficiency. Different modeling methods, time steps, order and the application of the quasi-Newton method all have a bearing on the efficiency of the solution. Finally, a comparison with an experiment in the free-pendulum simulation reveals the capability of this model to simulate dynamic problems with air resistance.

**Keywords:** space-time finite element method; absolute nodal coordinate formulation; cable element; constraint; dynamic solver

## 1. Introduction

Currently, flexible cable is wildly used in many fields, such as cable-driven parallel robots [1–3], submarine communication cables [4], high tension power lines [5] and some spacecrafts [6]. Flexible cable is anisotropic in mechanics. A flexible cable under tension can usually be simplified as a two-force bar on some occasions [7]. However, it undergoes a complex deformation and dynamics such as self-collision and self-winding when it turns into a slack cable. The low Young's modulus and the surrounding environment intensified this condition [8]. The flexibility of the cable should be considered in laying submarine cables and electric wiring. One of the most infamous cases without taking the flexibility of cable into consideration is the failure of the tethered satellite system TSS-1 of Italy in 1992. Therefore, the simulation of flexible cables is one of the most important problems in multibody dynamics.

With the development of multibody dynamics, two major modeling methods are used in multibody dynamic called floating frame reference formulation (FFRF) and absolute nodal coordinate formulation (ANCF). ANCF is proposed by Shabana [9]. It uses position

vector and the position vector gradient to describe the movement. Compared with FFRF, ANCF has better performance in large deformation [10] and rotation [11,12]. At first, the gradient vector is defined in the beam element [13], and extend to the plate element and solid element. Flexible cables are often stressed by tension and bending forces without torsion. Gerstmayr [14] proposed the cable element. With the improvement of accuracy and efficiency, it is appropriate to simulate the dynamic of flexible cables. This ANCF cable element is proposed by simplification from a three-dimensional beam. Gerstmayr [15], and Sugiyama [16] replaced the geometrical curvature with a material measure of curvature [17] for raising the calculation accuracy in the bending process. The stress continuity is $C^1$ at the middle of the cable, but it is $C^0$ at the boundary. This phenomenon will decrease the accuracy of stress between two elements. In view of this situation, Zhang [18] used high-order interpolation to improve the stress discontinuity in different element sections.

The discretization of the original ANCF is basically used in the space direction to transform a partial difference equation (PDE) into an ordinary difference equation (ODE). Many ODE solvers can be used for ANCF dynamic solutions, such as Newmark method [19], generalized $\alpha$ method [20] and Runge–Kutta method [21]. These methods are called half space-time finite element methods, which means the PDEs are discretized by finite element method in space and solved by ODE solvers. Half space-time discretization is not usually symplectic preserving. The energy convergence and long-term stability of the non-symplectic algorithm is worse than that of the symplectic preserving algorithm [22]. Moreover, half space-time discretization usually cannot use unstructured grids in the time dimension, which may cause a lack of flexibility in inconsistent time advancing for different regions [23].

A series of space-time finite element methods are proposed for good energy convergence and high efficiency. The first space-time finite element method was proposed by Argyris [24,25]. The early space-time finite element method used the Galerkin variation formulation [26], which makes the space-time meshing and asynchronous solution possible [27]. It also makes the meshing adapt to the time-varying boundary conditions [28,29], increased grid flexibility and computational efficiency. The Hamilton variation method improves the efficiency and long-term stability. Zhong [30] proposed a Hamiltonian FEM method in the time direction. It shows good energy conservation and solution efficiency. Gao [31–33] developed this method and extended it to the space-time finite element method. Then, Mergel [34] and Sánchez [35] extended it to Hamilton's law of variable action, so that work generated by non-conservative force can be calculated in natural variations. Additionally, the space and time directions are regarded the same in space-time finite element method. The position and velocity of one or more steps in the space-time finite element are presented as variables in discrete form and solved in explicit formulation. In this way, the PDE can be discretized into nonlinear equations without using ODE method. Space-time discretization simplifies the discrete scheme and also improves the flexibility and efficiency of the solution.

At this time, many algorithms to solve nonlinear equations such as Newton iteration algorithm and Quasi-Newton algorithm can be used to solve the dynamic problem in space-time discretization formulation [36]. The Newton iteration algorithm family is wildly used in solving nonlinear equations due to its second-order rate of convergence and its robust characteristic [37]. However, it needs to calculate the inverse of Jacobian matrix each iteration. The quasi-Newton algorithm can increase the efficiency via decrease inversions, and some of the quasi-Newton methods have superlinear convergence. Both methods are used in different aspects. Quasi-Newton methods should be preferred for compressible mechanics, whereas inexact Newton–Krylov methods should be preferred for incompressible problems [38]. In the quasi-Newton algorithm, Davidon–Fletcher–Powell (DFP) formula and Broyden family formula [39] are two ordinary formulas. Broyden methods can be further divided into a rank-1 method and rank-2 method according to the rank difference. A higher rank takes higher complexity and lower efficiency [40]. However, less inversion times do not necessarily mean that the computational efficiency will be improved. The

computational efficiency of the Newton method or quasi-Newton method is different for practical problems [41]. A lower matrix dimension and higher non-linearity will make the Newton method with better convergence have higher computational efficiency [42].

This article proposes a space-time absolute nodal coordinate formulation discretization method, which can convert ordinary ANCF cable elements into space-time ANCF cable elements. In Section 2, two kinds of space-time ANCF cable (SAC) elements are proposed with this method. Formulations of volume force, bending moment and air resistance are also given. A dynamic solver based on natural variation and two kinds of constraints are proposed. In Section 3, a single-end bending model is proposed to verify the accuracy of different elements with a static simulation of cantilever under end bending moment. An opposite velocity impact alone and vertical to the centerline of the cable is another example to show the efficiency of different elements and solvers. In Section 4, a dynamic simulation and an experiment of a free pendulum is shown to study the influence of air resistance.

## 2. Establishment of SAC Element

To generate spatiotemporal elements, it is important to distinguish between the space-time ANCF cable (SAC) element and the original ANCF cable (OAC) element. The main difference between the SAC and OAC elements lies in the shape function. When finite element method is used for discretization in both space and time domains, the shape function includes space and time parameters. The location vector and gradient vector can both affect the calculation of the displacement field. SAC introduces the gradient of material coordinate and its speed for kinematic description [43], which means that the generalized coordinates include both position vectors and velocity vectors.

The spatiotemporal generalized coordinate in the matrix format is given in

$$\mathbf{r} = \mathbf{S}(x,t)\mathbf{q}_m. \tag{1}$$

$\mathbf{r}$ represents the position coordinate of a random point on the neutral axis. $\mathbf{S}$ is the element shape function at one dimension, and $\mathbf{q}_m$ is the generalized coordinates and velocities. The material coordinate $x$ and time coordinate $t$ in the shape function are independent and the shape function and the number of degrees of freedom (DOFs) are determined by both the material coordinate component and the time direction component of the shape function.

The Lagrange method is a way to generate the shape function of a rectangular space-time element by multiplying the spatial and temporal shape function with the Lagrangian family base function. The algorithm is shown in

$$\mathbf{S}(\xi,\tau) = \mathbf{S}^X(\xi)\mathbf{S}^T(\tau). \tag{2}$$

where $\xi$ and $\tau$ are dimensionless coordinates represented in

$$\xi = \frac{x}{l}, \tau = \frac{t}{\Delta t}. \tag{3}$$

when using the Lagrange method, since the shape functions of time and space are independent of each other, it is possible to adjust the shape function in a single direction to satisfy the different continuity requirements between elements. Two cable elements based on different spatial shape functions are proposed for verifying the validity of this method. Berzeri and Shabana [44] proposed the first of the two spatial shape functions, which employs Hermitian interpolation and is expressed as

$$\mathbf{S}^X(\xi) = \begin{bmatrix} 1 - 3\xi^2 + 2\xi^3, & l(\xi - 2\xi^2 + \xi^3), & 3\xi^2 - 2\xi^3, & l(-\xi^2 + \xi^3) \end{bmatrix}. \tag{4}$$

Zhang [18] proposed the second of the two shape functions to ensure the continuity of stress in the section between different elements, which includes curvature terms as shown in

$$
\mathbf{S}^{X}(\xi) = \begin{bmatrix} -6\xi^5 + 15\xi^4 - 10\xi^3 + 1 \\ l\left(-3\xi^5 + 8\xi^4 - 6\xi^3 + \xi\right) \\ l^2\left(-\frac{1}{2}\xi^5 + \frac{3}{2}\xi^4 - \frac{3}{2}\xi^3 + \frac{1}{2}\xi^2\right) \\ 6\xi^5 - 15\xi^4 + 10\xi^3 \\ l\left(-3\xi^5 + 7\xi^4 - 4\xi^3\right) \\ l^2\left(\frac{1}{2}\xi^5 - \xi^4 + \frac{1}{2}\xi^3\right) \end{bmatrix}^T. \tag{5}
$$

The first of the two elements (SAC-2) employs four vectors to interpolate the shape function, including the position vector, gradient of material coordinate, velocity vector, and the velocity vector of gradient. The second of the two elements (SAC-3) incorporates two additional vectors, namely the curvature vector and its velocity, resulting in a total of six usage vectors. To ensure the temporal continuity, Hermitian interpolation is employed, requiring the use of shape functions. The shape function is expressed as

$$
\mathbf{S}^T(\tau) = \begin{bmatrix} 1 - 3\tau^2 + 2\tau^3, & \Delta t\left(\tau - 2\tau^2 + \tau^3\right), & 3\tau^2 - 2\tau^3, & \Delta t\left(-\tau^2 + \tau^3\right) \end{bmatrix}. \tag{6}
$$

The displacement field of an element can be expressed using two different shape functions, as shown in Table 1. Though their temporal component remains the same, their spatial components differ. With this method, two spatiotemporal finite elements are constructed, one ensuring spatiotemporal $C^1$ continuity (SAC-2) and the other ensuring $C^2$ continuity in the spatial direction and $C^1$ continuity in the time direction (SAC-3).

**Table 1.** The shape functions of different elements.

| Element No. | SFC [1]-x | SFC [1]-t |
|:---:|:---:|:---:|
| SAC-2 | Equation (4) | Equation (6) |
| SAC-3 | Equation (5) | Equation (6) |

[1] shape function component.

Due to the Euler–Bernoulli assumption, the cross-section of the beam is perpendicular to the neutral axis, and the section of cable is considered to be rigid. Under this assumption, it is considered that the strain on the cross-section is linear distributed with respect to the distance against the neutral axis. The strain of the cable can be generated with

$$
\varepsilon = \varepsilon_c + \kappa y. \tag{7}
$$

$\varepsilon_c$ is the centerline strain and can be directly calculated with position vectors $\mathbf{r}_x$ as shown in

$$
\varepsilon_c = \frac{1}{2}(\mathbf{r}_x \mathbf{r}_x - 1). \tag{8}
$$

The curvature term $\kappa$ is presented in

$$
\kappa = \frac{\left|\mathbf{r}_x \times \mathbf{r}_{xx}\right|}{\left|\mathbf{r}_x\right|^2}. \tag{9}
$$

The dynamic equation of the SAC element needs to be established using Hamilton's law of varying action to ensure the energy conservation properties of the element in a conservative system and its adaptability to non-conservative external forces [34]. The action of the element can be expressed as

$$
A^e = \int L^e dt = \int \left(T^e - U^e + \sum W^e\right) dt. \tag{10}
$$

Action $A^e$ is the time integral of Lagrange function $L^e$. $T^e$ is kinetic energy, $U^e$ is deformation energy and the additional work $W^e$ by external forces. $T^e$ and $U^e$ can be shown as

$$T^e = \frac{1}{2}\mathbf{q}_m^T \int_V \rho \mathbf{S}^T \cdot \mathbf{S} \mathrm{d}V \mathbf{q_m}. \tag{11}$$

$$U^e = \frac{1}{2}\int_V \left(EA\varepsilon_c^2 + EJ\kappa^2\right)\mathrm{d}V. \tag{12}$$

The external work $W^e$ has two components, $W_{vf}^e$ and $W_{sf}^e$, generated by equivalent nodal forces. These correspond to the work of the volume force and surface force, respectively, and can be expressed as integrals over their corresponding volumes or surfaces, as shown in

$$W_{vf}^e = \int_V \rho_A \mathbf{g} \cdot \mathbf{r} \mathrm{d}V. \tag{13}$$

$$W_{sf}^e = \int_S \mathbf{F} \cdot \mathbf{r} \mathrm{d}S. \tag{14}$$

In Equation (13), $\rho_A$ is the linear density of the cable. $\mathbf{g}$ is the volume force density and the $\mathbf{F}$ is the surface nodal force. Besides, the moment can be regarded as a special nodal force that can be calculated as additional work by integrating the moment and torsion angle. The work generated by the moment can be given in tensor form:

$$W_M = M_{ij}\theta_{ij}. \tag{15}$$

In Equation (15), $M_{ij}$ is the moment with tensor expression. The bending angle $\theta_{ij}$ is obtained as the arctangent of the material coordinate gradient in the SAC expression, as shown in

$$\theta_{ij} = \tan^{-1}\left(\frac{r_{i,x}}{r_{j,x}}\right). \tag{16}$$

$r_{i,x}$ is the component if the position gradient $\mathbf{r}_x$ at i direction. If the bending angle lies in the x-y plane, then $i = 1$ and $j = 2$.

Air resistance is a typical case to show the influence of resistance and can be regarded as a kind of surface force, while the plane of action is the windward section. The value of air resistance related to the resistance parameter, the density of air, the area of the windward section and the velocity of the test node on the neutral axis and the direction is opposite to velocity. The work made by air resistance is shown in

$$W_{far}^e = \int_x -\frac{C\rho_{Air}S_w\mathbf{v}\cdot\mathbf{v}}{2}\cdot\mathbf{r}\cdot\frac{\mathbf{v}}{|\mathbf{v}|}\mathrm{d}x. \tag{17}$$

In Equation (17), $C$ is the drag coefficient, and it is related to the Reynold number, the shape of the windward and some other conditions. $\rho_{Air}$ is the density of air. $S_w$ is the surface area of the windward. $\mathbf{v}$ is the velocity vector.

Equations (11)–(13) and (17) involve numerical integration, particularly Gauss integration in the spatial domain. The action given in Equation (10) is also evaluated using Gauss integration in the time domain. The number of Gauss integration points depends on the shape function and the calculation of kinetic energy, potential energy, and work. This process is similar to selecting Gauss integration nodes in a static planar problem. However, unlike in planar static problems, the selection of spatiotemporal Gauss integration nodes replaces the y-axis with the time axis, as shown in Figure 1.

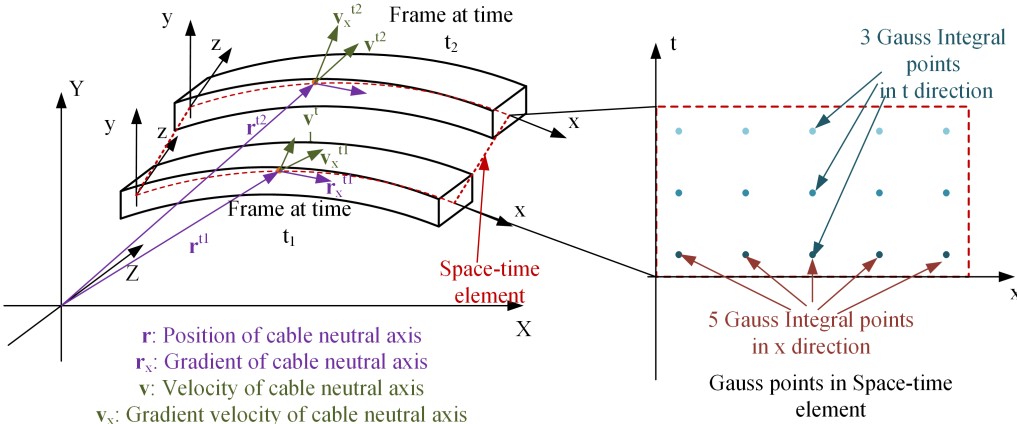

**Figure 1.** The frame of SAC element and the distribution of Gauss point.

Each element must satisfy the Courant–Friedrichs–Lewy (CFL) condition to ensure a convergent solution when using a rectangular space-time mesh. The CFL condition for longitudinal waves is given in

$$C = \sqrt{\frac{E\Delta t}{\rho \Delta x}} \leq \frac{1}{\sqrt{2}}. \tag{18}$$

This means that when the mesh in the spatial direction is determined, the maximum time step is also determined. However, this condition does not restrict the assembly in the time direction, which can circumvent the CFL condition and enable long-step solutions.

In this section, the establishment method from OAC element to SAC element is proposed and two types of SAC element are constructed with different spatial shape function. The integral of Lagrangian function and action by kinetic energy, elastic energy and external force are given. The bending moment and air resistance formulation are given. The numerical integration method point is also discussed at last.

## 3. Element Assembly and the Solver Construction

In the preceding section, the establishment of SAC elements is discussed. The global action can be expressed as the sum of element actions. In this chapter, the assembly of SAC elements and the expression of the global action will be discussed.

The beginning and the ending frame of a space-time assembly should be study alone in some discretization method due to the particularity. So, the space-time nodes are divided into input time node ($t = t_a$), intermediate time node ($t = t_i$) and output time node ($t = t_b$) according to their time. The DOFs have time property, which subject to nodes and be divided. Node and its DOFs have the same definition in time. The Hamilton's law of varying action is shown as

$$\delta A = \left[ \mathbf{H^1} \left( \sum_{E=1}^{n} \frac{\partial A^E}{\partial \mathbf{q}} + \mathbf{M}^{ab} \mathbf{q} \right) + \mathbf{H^2} \left( \mathbf{q}^G - \mathbf{q}^{G0} \right) \right] \delta \mathbf{q}_e^G + \left[ \int \boldsymbol{\lambda}^T \frac{\partial \mathbf{R}}{\partial \mathbf{q}_e^G} dt \right] \delta \mathbf{q}_e^G. \tag{19}$$

$\mathbf{q}_e^G$ is composed by the global generalized coordinate $\mathbf{q}_e$ with Lagrange multipliers $\boldsymbol{\lambda}$. $\mathbf{q}^{G0}$ is the initial value of global generalized coordinate. $A^E$ is the derivation of element action with respect to $\mathbf{q}_e$. $\mathbf{R}$ are constraints constructed by the Lagrange multiplier method. The assembly of action is similar to original assembly of FEM, but there are three main differences, which are choices of $\mathbf{M}^{ab}$, $\mathbf{H^1}$ and $\mathbf{H^2}$. These three parameters are related to the input–output time node and the moment expression. That is why different temporal nodes are defined at the beginning of this section. The moment expression proposed by Marsden and West and the pseudo-moment expression depicted the relationship between the position and velocity in input time and output time. Refer to the study of convergence,

stability and symplectic preserving characteristics of the solver by Mergel [34]. It performs well in most of aspects to use the P2 formula to construct the solver, in which its $\mathbf{M}^{ab}$ is given as

$$\mathbf{M}^{ab} = diag\left(\begin{array}{ccc} \mathbf{M}^a & \mathbf{0} & -\mathbf{M}^b \end{array}\right). \tag{20}$$

$\mathbf{M}^a$ is the mass matrix at time $t_a$ and $\mathbf{M}^b$ is the mass matrix at time $t_b$. The element mass matrix at a random time slice of SAC is equal to the element mass matrix of OAC. Moreover, both $\mathbf{H}^1$ and $\mathbf{H}^2$ are given as

$$\left\{ \begin{array}{l} \mathbf{H}^1 = diag\left(\begin{array}{ccccccc} \mathbf{I}_{\mathbf{e}_a} & \mathbf{I}_{\mathbf{e}_i} & \mathbf{I}_{\mathbf{e}_b} & \mathbf{0}_{\mathbf{v}_a} & \mathbf{I}_{\mathbf{v}_i} & \mathbf{0}_{\mathbf{v}_b} & \mathbf{0}_\lambda \end{array}\right) \\ \mathbf{H}^2 = diag\left(\begin{array}{ccccccc} \mathbf{0}_{\mathbf{e}_a} & \mathbf{0}_{\mathbf{e}_i} & \mathbf{0}_{\mathbf{e}_b} & \mathbf{I}_{\mathbf{v}_a} & \mathbf{0}_{\mathbf{v}_i} & \mathbf{I}_{\mathbf{v}_b} & \mathbf{0}_\lambda \end{array}\right) \end{array} \right. . \tag{21}$$

$\mathbf{e}_a$, $\mathbf{e}_i$, and $\mathbf{e}_b$ are the position at input, intermediate and output time. $\mathbf{v}_a$, $\mathbf{v}_i$, and $\mathbf{v}_b$ are the velocities at input, intermediate and output time. The discretized dynamic functions are presented with the following three equations:

$$\left\{ \begin{array}{l} \frac{\partial A}{\partial \mathbf{e}_a} + \mathbf{M}_a \mathbf{v}_a = \mathbf{0} \\ \frac{\partial A}{\partial \mathbf{e}_i} = \mathbf{0} \\ \frac{\partial A}{\partial \mathbf{e}_b} - \mathbf{M}_b \mathbf{v}_b = \mathbf{0} \\ \frac{\partial A}{\partial \mathbf{v}_i} = \mathbf{0} \end{array} \right. . \tag{22}$$

$$\left\{ \begin{array}{l} \mathbf{e}_a - \mathbf{e}_0 = \mathbf{0} \\ \mathbf{v}_a - \mathbf{v}_0 = \mathbf{0} \end{array} \right. . \tag{23}$$

$$\mathbf{R} = \mathbf{0}. \tag{24}$$

　　　Notably, two types of constraint equations are proposed in this paper, which are called replaceable constraint (RC) in Equation (23) and supplementary constraint (SC) in Equation (24) separately. These two constraints can replace each other between the two equations under the solver formulation. In the absence of SCs, RCs usually appear as constraints of initial value conditions. However, the form of RCs is not fixed. In P2 solver, RCs could not include the velocity term, which will lead to the matrix singularity, such as the fixed end constraint and hinged constraint. These constraints require both the position and velocity constant in P2 solver. When the SCs contain the velocity term, it is necessary to convert some position constraints of the RCs to the velocity constraints in the SCs. That is, the same number of position constraints in SCs are changed into RCs, and the same number of nodal velocity SCs are replaced with position constraints of RCs. The total number of constraints remains the same. This process is called constraint conversion. The SCs will not contain any terms that include velocity at the input and output time, while the RCs may contain both velocity items and position items. Moreover, RCs only replace the velocity at the input and output time. The constraints of intermediate time are calculated as SCs. This characteristic is defined due to the particularity of input and output time.

　　　It takes the fixed end constraint as an example in Figure 2 to specify the constraint conversion process. If a cable without constraint as shown in Figure 2a, there are no SCs in this formulation, and all the constraints are the initial conditions and set in RCs. Figure 2b shows a cable with a cantilever constraint. After meshing this cable in four elements, the position and velocity keep constant in nodes 1, 4, and 7. So, all the DOFs including position and time subject to the nodes 1, 2, 3 are set as RCs, and the nodes 4 and 7 are set as SCs. The SCs in nodal 7 are needed to start the constraint conversion process by replacing all the velocity constraints to SCs. When the position constraints are finished, the RCs are all the position constraints, the SCs are the velocity constraints of the input and output time and both are constraints of the intermediate time.

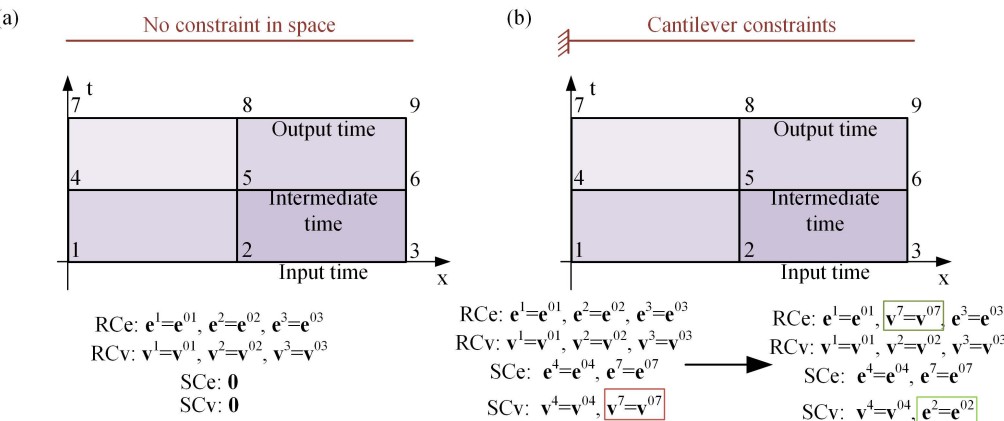

**Figure 2.** (**a**) RC and SC constraints in no-constraint cable. (**b**) RC and SC constraints in cantilever cable and the constraint conversion process.

The Equations (22)–(24) are nonlinear functions, which can be solved with Newton–Raphson (N-R) algorithm and Broyden rank 1 (BR1) algorithm. The format of N-R algorithm is shown as

$$\mathbf{X}^{i+1} = \mathbf{X}^i - \mathbf{F}_q{}^{-1}\mathbf{F}^i. \tag{25}$$

$\mathbf{X}$ is the variables, and $\mathbf{F}$ is nonlinear equations. $\mathbf{F}_{,q}$ is the Jacobian matrix of the nonlinear equations. $i$ is the times of iteration. The terminal condition of N-R algorithm is the 2-norm of residual error less than tolerance. This relation is shown as

$$|\mathbf{X}^{i+1} - \mathbf{X}^i| < \mathbf{R}_0. \tag{26}$$

The BR1 algorithm is a kind of quasi-Newton method and it is shown with

$$\begin{cases} \mathbf{X}^{i+1} = \mathbf{X}^i - \mathbf{B}^i\mathbf{F}^i \\ \mathbf{y}^i = \mathbf{F}^{i+1} - \mathbf{F}^i \\ \mathbf{B}^{i+1} = \mathbf{B}^i + \frac{(\mathbf{s}^i - \mathbf{B}^i\mathbf{y}^i)\mathbf{s}^i\mathbf{X}^i}{\mathbf{X}^{iT}\mathbf{B}^i\mathbf{y}^i} \end{cases}. \tag{27}$$

Both N-R algorithm and BR1 method need to calculate the Jacobian matrix at the initial time. The initial matrix $\mathbf{B^0}$ is the inverse of Jacobian matrix $\mathbf{F}^0_{,q}$. However, the BR1 method only calculates the inverse of the Jacobian matrix one time different from N-R algorithm, which calculates the inverse of the Jacobian matrix at each iteration. Hence, the BR1 method has a higher efficiency in one iteration than N-R algorithm.

In this chapter, a solution method based on natural variation is used to assemble the space-time discrete dynamic formulation. The mass matrix in different times and the P2 formula are used to construct this solver. Two types of constraints and the constraint conversion method are proposed. Newton–Raphson algorithm and BR1 method are introduced to solve the dynamic functions.

## 4. Simulation and Verification

A static simulation and a dynamic simulation are presented to show the differences between SAC-2 and SAC-3. The accumulation difference in two different elements are studied with the end bending moment in a static simulation. In the dynamic simulation, the conservation of total energy is discussed at first. The continuity of stress and the efficiency of different modeling methods are also discussed.

### 4.1. Static Simulation

To verify the accuracy of the elements and constraints derived, a static example of a cantilever beam subject to the single-end bending moment is presented. In theory, the

solution of the static equation is independent of time, and the time integral of the space-time finite element method reduces to a constant value. Moreover, the presence of temporal Gauss integral points extends the solution time, which takes more time than using the principle of virtual displacement. Using the space-time finite element method to solve the static problem can not only confirm the accuracy of the different elements, but also provide a good example to demonstrate the constraint conversion mentioned previously.

In this section, a one-meter PVC cantilever is used as an example. The cantilever's properties are listed in Table 2, with the fixed end referred to as End A and the free end as End B. The cantilever is discretized with 5 SAC-2 elements and 10 SAC-3 elements in space. A constant bending moment $M$ is applied at End B, and the effect of gravity is neglected throughout the bending process. The relationship between the bending moment and the rotation angle at End B is presented as

$$M = \frac{\lambda \pi E I}{l}. \tag{28}$$

The $E$ is the Young's modulus of the cantilever. $I$ is the moment of inertia. $l$ is the length of the cantilever. When $\lambda = 1$, the constant bending moment can make the cantilever into a semicircle, and it can also make the cantilever into a circle theoretically when $\lambda = 2$.

**Table 2.** Material properties of static simulation.

| Material Properties | Value |
|---|---|
| Elastic modulus (GPa) | 3.4 |
| Density (kg/m$^3$) | 1380 |
| Moment inertia (m$^4$) | $1.25 \times 10^{-13}$ |
| Area (m$^2$) | $3 \times 10^{-6}$ |

The bending situation is shown in Figure 3, and the deviation in the X direction is shown in Table 3. Both SAC-2 and SAC-3 can bend into a semicircle under the application of a bending moment using 10 elements. However, SAC-3 can bend more accurately into a circle with less error compared to SAC-2. The horizontal error of 5 SAC-3 elements is even less than that of 10 SAC-2 elements, indicating that SAC-3 elements have higher accuracy in bending problems.

**Table 3.** The horizontal error of ending point with moment.

| Element Type and Constraint | 5 Elements | | 10 Elements | |
|---|---|---|---|---|
| | Semicircle | Circle | Semicircle | Circle |
| SAC-2 | 65.31 | $-197.3$ | 0.400 | 18.30 |
| SAC-3 without $r_{xx}$ constraint | 0.172 | $1.380 \times 10^{-3}$ | $-4.622 \times 10^{-4}$ | $2.636 \times 10^{-6}$ |
| SAC-3 with $r_{xx}$ constraint | No data | No data | 21.17 | 0.063 |

However, the constraints of SAC-3 present certain issues due to the absence of physical meaning of high-order gradients. In the static solution, all the velocity-related parameters need to be set as replaceable constraints with a constraint conversion method, and the position vector and position vector gradient at End A are also set as supplementary constraints in both the input and output time to restrict the movement and rotation of End A. If the constraint of the second-order gradient is set, the error increases, as shown in Table 3. The position vector of the analytic solution at the beginning and end of the bending process is compared, and the expression of the analytic solution is presented as

$$\begin{cases} \mathbf{r}_{ts} = \begin{bmatrix} x, & 0, & 0 \end{bmatrix} \\ \mathbf{r}_{tf} = \begin{bmatrix} y_0 \cos\theta, & y_0 \sin\theta + y_0, & 0 \end{bmatrix} \end{cases}. \tag{29}$$

At the beginning, the neutral axis is a segment with a material coordinate to coincide with the global coordinate. It becomes a circle with a radius of $y_0$ and the center is $\begin{bmatrix} 0 & y_0 & 0 \end{bmatrix}$ at the end. $y_0$ is related to the length of the cable as

$$2\pi y_0 = l. \tag{30}$$

The second derivative of Equation (29) is presented as

$$\begin{cases} \frac{\partial^2 \mathbf{r}_{ts}}{\partial x^2} = \begin{bmatrix} 0, & 0, & 0 \end{bmatrix} \\ \frac{\partial^2 \mathbf{r}_{tf}}{\partial x^2} = \begin{bmatrix} 0, & \frac{\cos 2\theta}{y_0 \sin^2\theta}, & 0 \end{bmatrix} \end{cases}. \tag{31}$$

The vector of the second derivative is $\begin{bmatrix} 0, & 0, & 0 \end{bmatrix}$ at origin point at the beginning and $\begin{bmatrix} 0, & 2\pi, & 0 \end{bmatrix}$ at origin point at the end, the error of which is 1.15% with the simulation by SAC-3. So, in the fixed end bending simulation, the second-derivative should not be fixed.

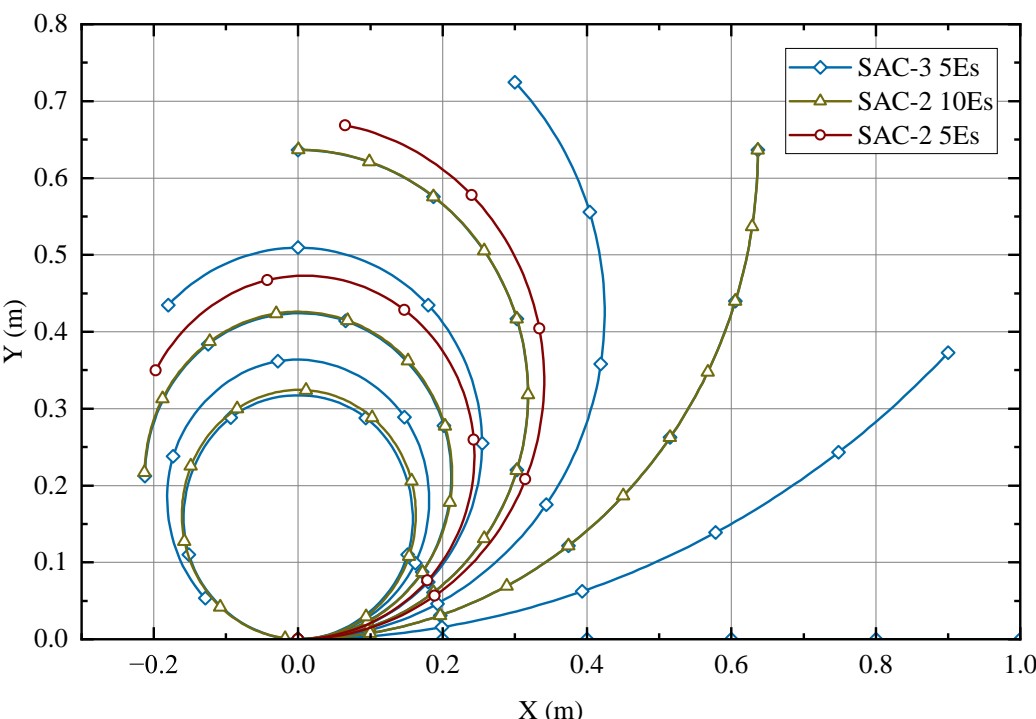

**Figure 3.** The circle-bending example with 5 and 10 SAC-2 elements and 5 SAC-3 elements.

### 4.2. Double-Ended Velocity Impact Simulation and Verification

The example of double-ended velocity impact primarily demonstrates the energy convergence of the space-time ANCF formulation. A pair of symmetric velocity vectors are applied to both ends of the cable, and no constraints are imposed on the cable. The simulation involves a 4-meter PVC cable with section and material properties listed in Table 2. The cable is modeled using 8 SAC-2 and SAC-3 elements separately to observe the distribution of stress waves at different times. The application point and velocity vector of the symmetric velocity are presented in Table 4. This model is designed for simplicity, with only one layer of SAC elements and a time difference of $\Delta t$ between the two points.

**Table 4.** The initial condition of two examples.

| No. of Examples | Initial Condition of A End | | Initial Condition of B End | |
|---|---|---|---|---|
| | Position Vector | Velocity Vector | Position Vector | Velocity Vector |
| No.1 | $\begin{bmatrix} -2, & 0, & 0 \end{bmatrix}$ | $\begin{bmatrix} 10, & 0, & 0 \end{bmatrix}$ | $\begin{bmatrix} 2, & 0, & 0 \end{bmatrix}$ | $\begin{bmatrix} -10, & 0, & 0 \end{bmatrix}$ |
| No.2 | $\begin{bmatrix} -2, & 0, & 0 \end{bmatrix}$ | $\begin{bmatrix} 0, & 10, & 0 \end{bmatrix}$ | $\begin{bmatrix} 2, & 0, & 0 \end{bmatrix}$ | $\begin{bmatrix} 0, & -10, & 0 \end{bmatrix}$ |

In example-1, the response to the opposite velocity impact is illustrated in Figure 4. This velocity impact excites a compressive stress wave along the material coordinate. These two waves have an affected region of approximately 0.5 m and transmit at a longitudinal wave velocity of 1480 m/s, which is the reciprocal of the slope of the characteristic line. They then add at the center of the cable and cause a maximum absolute stress of 23.5 MPa. In the absence of any changes in material properties, the waves transmit without interaction until they reach the boundary. At the free boundary, reflection occurs and the compressive stress changes into tensile stress. The subsequent wave transmission is similar to the compressive stress wave, except for the change in stress direction.

The continuity of stress is not the same when different elements are used for discretization. Sharp stress distribution occurs at the boundary between two SAC-2 elements, as shown in Figure 4a,b. In contrast, the stress distribution at the boundary is smooth with SAC-3 element discretization. The reason for this is that 5th-order interpolation guarantees $C^1$ stress continuity at the boundary, while 3rd-order interpolation only guarantees $C^0$ stress continuity at the boundary. Therefore, high-order interpolation can improve stress continuity, which enhances the accuracy of elastic stress wave calculations between two elements.

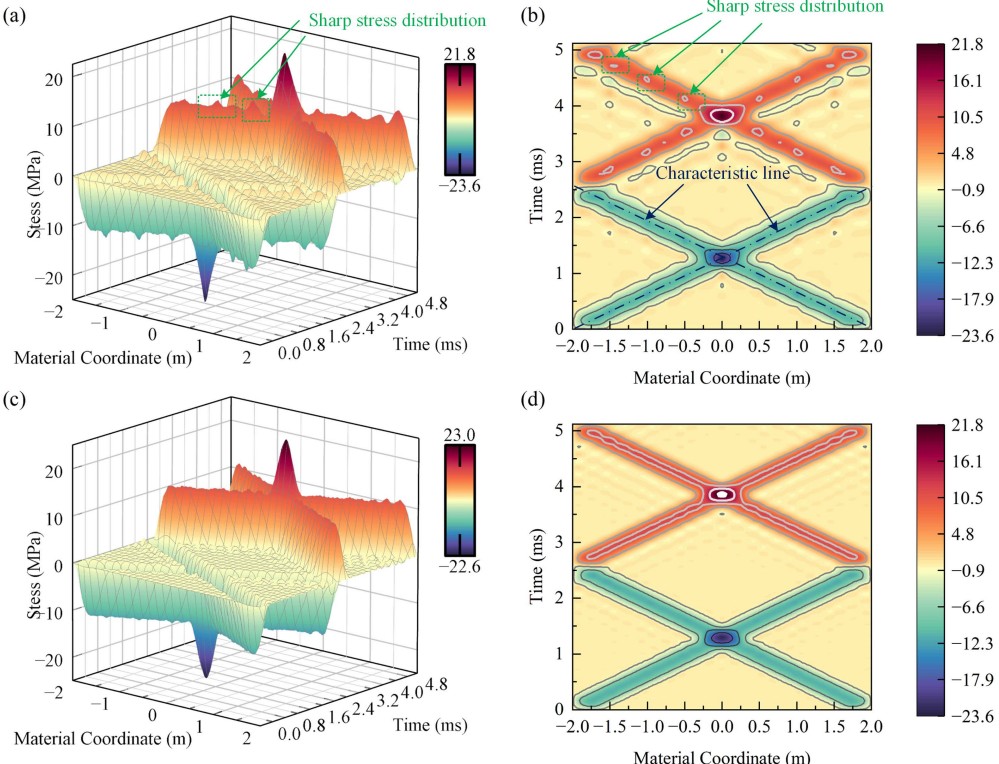

**Figure 4.** The transfer of longitude stress waves along the cable. (**a**) The 3D space-time stress distribution figure discretized by SAC-2 element. (**b**) The contour line figure of (**a**). (**c**) The 3D space-time stress distribution figure discretized by SAC-3 element. (**d**) The contour line figure of (**c**).

In example-2, the response to asymmetric velocity is illustrated in Figure 5. The motion is centrally symmetric, with the bending and rotation of the cable being the primary movements due to elasticity over 1.8 s. The conservation of the middle point position of the cable and total energy is shown in Figure 5a. Without any restrictions, the middle point of the cable remains at the original position. The variation of potential energy, kinetic energy, and the summary of potential and kinetic energy over time are presented in Figure 5b. The kinetic and potential energy exhibit periodicity, but their summary remains nearly constant over time. These two results illustrate the conservation of the space-time ANCF method.

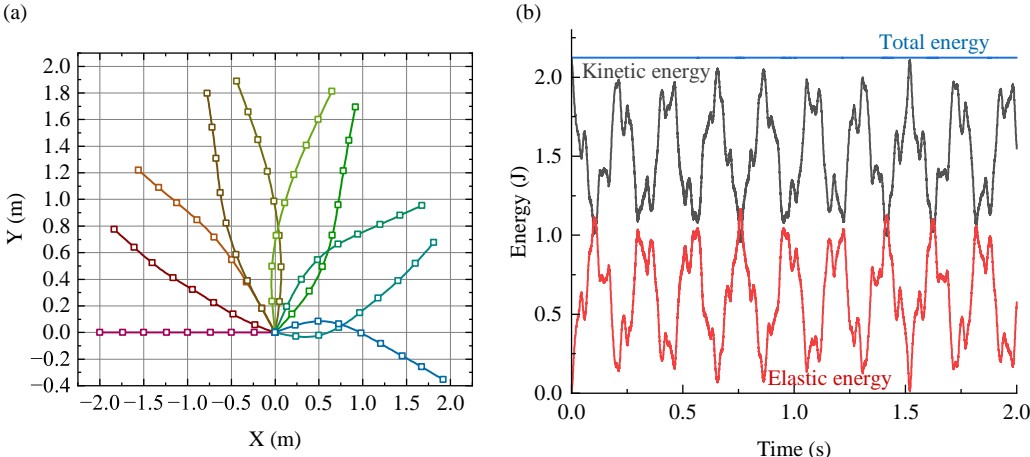

**Figure 5.** The dynamic simulation of example-2. (**a**) The spatial state of motion of the left end rope at the center point. (**b**) The energy conservation in double-ended shock issues.

### 4.3. The Efficiency of Space-Time ANCF Method

The studies above focus on factors that influence the accuracy of space-time element. Moreover, the order of interpolation, the size of element in space and time direction, the assembly method and different solve method will also influence the solution efficiency. The convergence and efficiency are discussed as follows.

The total time cost of a specific space-time ANCF method can be calculated with

$$t_{Total} = \left[ \underbrace{(n_E \times t_E)}_{Part\,A} + \underbrace{(t_A + t_S)}_{Part\,B} \right] \times n_I \times n_s. \tag{32}$$

$n_E$ is the number of element. $t_E$ is the establishment time of one element. $t_A$ is the assembly time in each iteration. $t_S$ is the solution time in each iteration. $n_I$ is the iteration times. $n_S$ is the number of calculation steps.

Examples based on example-1 in Section 4.2 are shown in Table 5. The purpose of these examples is to study the efficiency of the different parameters in Equation (32). The maximum deviation is used as the termination condition. The spending time, residual error, and iteration times of these examples are shown in Figure 6.

In the case where SAC-2 elements were used uniformly, refinement was carried out in both the temporal and spatial directions. The refinement interval in the temporal direction was 20 ms, and in the spatial direction it was 3 m. The Newton–Raphson method and Broyden Rank 1 method were used to calculate different grids. The system is conservative, so the energy conservation property can be used as the method to measure computational errors. The total energy of the flexible cable, including kinetic energy and elastic potential energy, is calculated at each solution step. The 1-norm of the difference between the energy at each step and the initial energy was used as the evaluation criterion to measure the error. The result is shown in Table 6.

**Table 5.** Examples to study the influence on efficiency.

| Example No. | Element Type | Mesh Method [1] | Steps | Solver | Integral Points [1] |
|---|---|---|---|---|---|
| No.1 | SAC-2 | $8 \times 1$ | 400 | N-R | $3 \times 3$ |
| No.2 | SAC-2 | $8 \times 1$ | 400 | BR1 | $3 \times 3$ |
| No.3 | SAC-2 | $8 \times 1$ | 400 | N-R | $6 \times 3$ |
| No.4 | SAC-2 | $4 \times 1$ | 200 | N-R | $3 \times 3$ |
| No.5 | SAC-2 | $4 \times 2$ | 100 | N-R | $3 \times 3$ |
| No.6 | SAC-2 | $2 \times 1$ | 100 | N-R | $3 \times 3$ |
| No.7 | SAC-3 | $4 \times 1$ | 200 | N-R | $5 \times 3$ |
| No.8 | SAC-3 | $4 \times 1$ | 200 | BR1 | $5 \times 3$ |
| No.9 | SAC-3 | $4 \times 1$ | 200 | N-R | $5 \times 6$ |

[1] The element distribution in Space $\times$ Time direction.

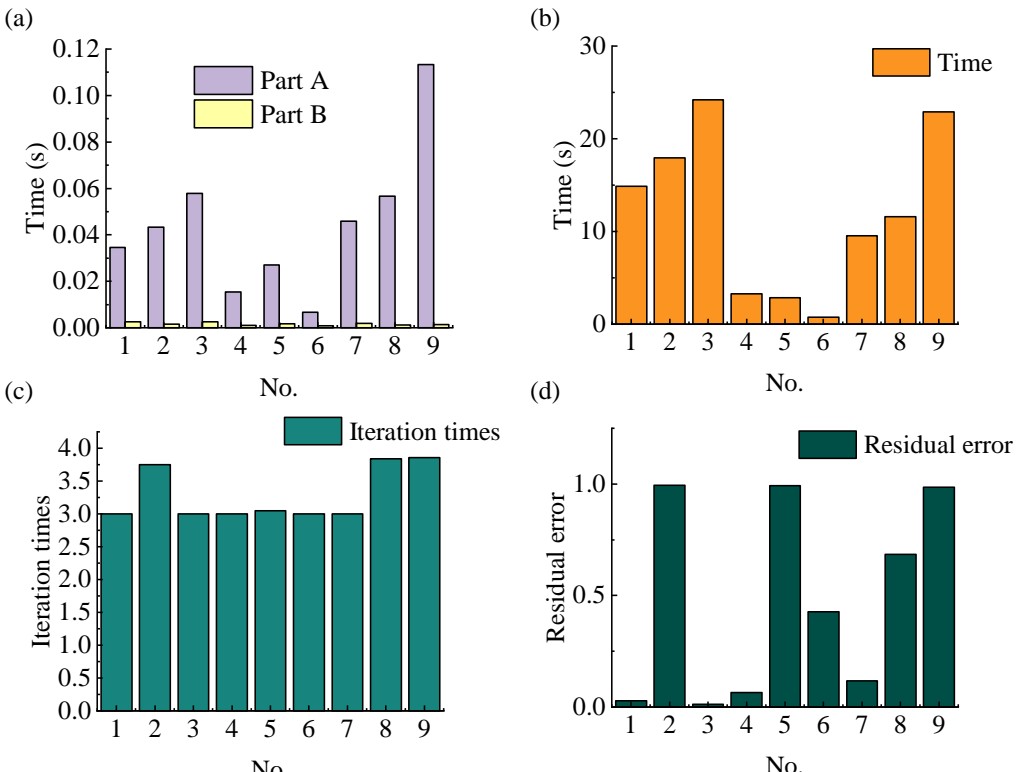

**Figure 6.** (**a**) The spending time by calculation in part A and part B of Equation (32). (**b**) Total spending times in different examples. (**c**) The average iteration times during calculation in different examples. (**d**) The residual error in different examples.

**Table 6.** Grid refinement test for the double-end impact on different grid types.

| Scheme | Grid | $L_1$ | Order |
|---|---|---|---|
| Newton-Raphson | 4TEs $\times$ 2SEs | 0.0040 | - |
| | 4TEs $\times$ 4SEs | 0.0032 | 0.3254 |
| | 8TEs $\times$ 2SEs | $2.66 \times 10^{-4}$ | 3.9099 |
| | 8TEs $\times$ 4SEs | $2.00 \times 10^{-4}$ | 4.2576 |
| Broyden rank 1 | 4TEs$\times$2SEs | 0.0040 | - |
| | 4TEs $\times$ 4SEs | 0.0032 | 0.3254 |
| | 8TEs $\times$ 2SEs | $2.66 \times 10^{-4}$ | 3.9099 |
| | 8TEs $\times$ 4SEs | $2.00 \times 10^{-4}$ | 4.2576 |

The impact of different refinement methods on accuracy is displayed. It can be seen from the table that spatial refinement has a smaller impact on the improvement of the

order of accuracy compared to temporal refinement, and the effect of mixed refinement is the best. Under this refinement method, N-R iteration method and BR1 iteration method have the same convergence accuracy when obtaining a convergent solution. This indicates that the choice of iteration method has little influence on the solution accuracy in this discrete format, and the difference between the two mainly lies in the convergence rate and convergence domain.

There are four major parameters that can affect calculation efficiency: the time step of the element, the interpolation order in space and time directions, the assembly in the time direction, and the solution method. The time step is studied first.

A longer time step results in fewer calculation steps. However, the time step is restricted by the CFL condition, which corresponds to the size in the space and time directions. One of the advantages of ANCF is that it can use fewer elements to simulate complex movements [18]. It can also use a larger time step by meshing with a coarse grid in space without sacrificing much accuracy. This advantage is also evident in the SAC-3 element, which uses 1.5 times the degrees of freedom (DOFs) compared to SAC-2 to achieve twice the accuracy. This is beneficial in reducing the number of elements to achieve high efficiency in certain situations. Examples No. 1, 4, and 6 of Table 5 show a decrease in time cost by using a coarse grid, but the residual error increases, which may require more iteration times for convergence.

The interpolation order affects the calculation time by changing the number of Gauss integral points. The exact integral order is usually the minimum order required to guarantee energy convergence. More Gauss points can be used to increase accuracy under strong nonlinearity. In the space-time ANCF method, the Gauss nodes are included in the space and time directions. The comparison of Examples No. 1 and 3 in Table 5 shows the influence of increasing the spatial Gauss nodes. Example No. 3 has the minimum residual error, but also the longest calculation time. In contrast, the increase in temporal Gauss nodes in Examples No. 7 and 9 did not show any advantage in efficiency. This implies that the spatial Gauss nodes can simultaneously increase convergence and decrease efficiency.

Assembly in the time direction is another way to reduce the number of calculation steps, and it is not restricted by the CFL condition. Example No.5, a double-layer model, is compared with Example No. 4, a one-layer model, in Table 5. Although the number of calculation steps in Example No. 5 is smaller than that in Example No. 4, the total element number remains the same, and the convergence becomes worse with a high maximum residual error and more average iteration times. The total calculation time is the same because the major time cost is in Part A, where the calculation complexity is much higher than in Part B.

The choice of different nonlinear dynamic equation solvers also affects the solution time of Part B. The Newton–Raphson iteration has a second-order convergence rate, and the total iteration loops around 3 times, but it needs to calculate the inverse of the Jacobian matrix each time. The inverse of a matrix has a calculation complexity of $O(n^2)$, so its Jacobian matrix is directly proportional to the square of the calculation time. The quasi-Newton method can only calculate the inverse once, but the convergence rate is only linear or superlinear. Examples No. 2 and 8 in Table 5 use the BR1 method to reduce the solution time of Part B, but the iteration loops increase due to the low convergence rate. The total calculation time is slightly higher than that of the Newton–Raphson method in this example.

The convergence history of the Newton iteration is shown in Figure 7. The calculation results of the convergence rate (CR) in Table 7 show that in seven of the nine examples where Newton–Raphson was used, super-linear convergence was observed. However, the two Broyden rank-1 methods showed less than super-linear convergence. This is consistent with the situation where Newton–Raphson has fewer iterations.

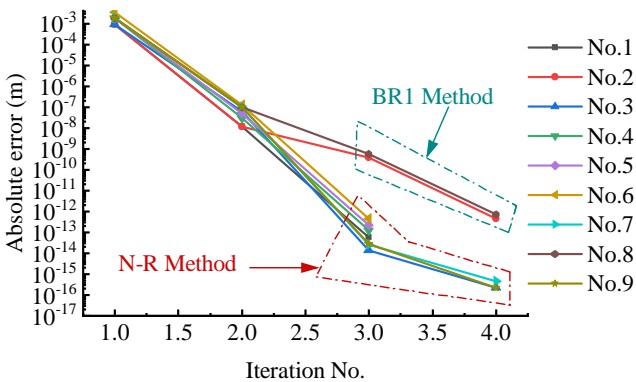

**Figure 7.** The convergence history of 9 examples.

**Table 7.** The convergence rate (CR) of 9 examples.

| No. | 1 | 2 | 3 | 4 | 5 | 6 | 7 | 8 | 9 |
|-----|-----|-----|-----|-----|-----|-----|-----|-----|-----|
| CR | 1.089 | 0.3040 | 1.592 | 1.125 | 1.171 | 1.219 | 1.538 | 0.524 | 1.526 |

The summary of these relationships is presented in Table 8. "+" indicates an increase, "−" indicates a decrease, and "/" represents no clear relationship between these two methods. Both longer time steps and time direction assembly can significantly improve efficiency without sacrificing much accuracy.

**Table 8.** Four efficiency increase methods and their effect on efficiency relevant parameters.

| Methods | Element No. | ECT [1] | Solution Time | Steps | ITC [2] | Accuracy |
|---------|-------------|---------|---------------|-------|---------|----------|
| Longer steps | −− | / | −− | − | / | − |
| Higher order | − | ++ | − | / | / | + |
| Time direction assembly | / | / | + | −− | / | / |
| Quasi-Newton method | / | / | − | / | + | / |

[1] Spending time in element calculation. [2] The iteration loops until convergence.

In this chapter, three different simulation models are proposed for verifacation. The static simulation shows the static solution ability of P2 solver, and the validity of the constraint conversion is also been verified. The double-ended velocity impact simulation shows that SAC-3 has a smoother stress distribution than SAC-2. Both elements can be used to solve dynamic problems, which shows the correctness of the element establishment. Longer steps and time direction assembly can improve efficiency better in four major parameters in the efficiency study. The assembly method and the element with better convergence would be the next research direction.

## 5. The Simulation and Experimental Verification of the Free Flexible Pendulum

The free flexible pendulum serves as a common illustration of ANCF cable for precise validation. This chapter employs a simulation and an experiment to authenticate the two categories of SAC element with consideration to air resistance. For the simulation and experiment, a 0.5-meter gutta-percha cable is utilized in the free-pendulum setup. The characteristics of this cable are presented in Table 9.

**Table 9.** Material properties of rubber (gutta-percha).

| Material Properties | Value |
|---|---|
| Elastic modulus (MPa) | 0.91 |
| Density (kg/m$^3$) | 1202 |
| Moment inertia (m$^4$) | $9.15 \times 10^{-12}$ |
| Area (m$^2$) | $1.075 \times 10^{-5}$ |

Previous research has often disregarded air resistance in free pendulum simulations to demonstrate energy convergence. However, in reality, air resistance significantly impacts free pendulum behavior. To illustrate this effect, a free pendulum is considered with air resistance at 0.68 s, using the same initial conditions as described above. At this point, the flexible pendulum has nearly completed half a period, and it is a defining moment due to its proximity to the minimum kinetic energy. The impact of air resistance on the flexible pendulum during the first half-period is primarily complete.

Figure 8 displays the flexible pendulum at 0.68 s with varying air densities. As the drag coefficient increases, the height of the pendulum globally decreases. This indicates the influence of air resistance discussed earlier. To compare with experimental results, air resistance must be taken into account. In this simulation, the drag coefficient is established as 1.5 based on the section's shape and the Reynolds number.

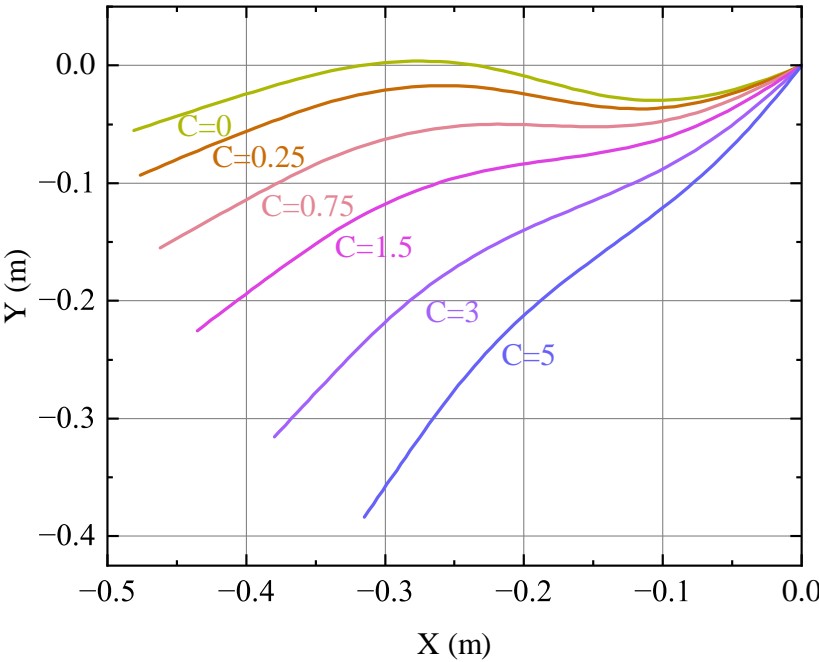

**Figure 8.** The simulation under different air resistances at $t = 0.68$ s.

The experimental setup is depicted in Figure 9, consisting of a frame structure, axis, shaft, flexible cable, and image acquisition equipment. The axis is made of Teflon due to its low density and self-lubrication properties. A green screen is positioned behind the frame structure to facilitate image processing. The Olympus Stylus 1 s camera and its components are used for image acquisition. The cross-section size is measured using a sliding gauge, and the cable's mass is determined using the YP202A electronic balance. The cable's density and Young's modulus are then calculated with successive minus method.

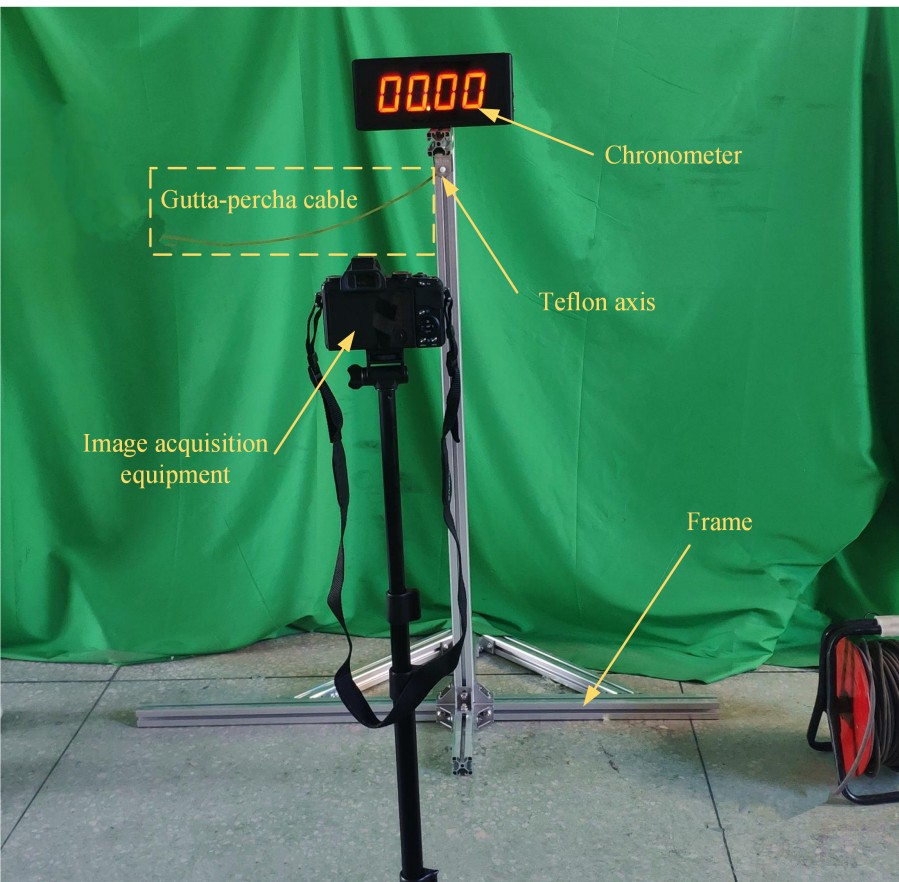

**Figure 9.** The structure of experimental facilities.

In this scenario, the cable is pinned at one end, referred to as End A, and left unconstrained at the other end, known as End B. The cable is placed horizontally and released at the start of the experiment. The camera takes pictures with a 1/800 s shutter duration until the cable comes to a halt. Seven images were obtained within 0.79 s, with Image 1 showing the state of the cable before it was released, which can be considered as the initial moment when the cable began to fall. The other six images were taken at different times between 0.19 and 0.79 s. By analyzing Image 1, the initial motion state of the cable was determined. The cable was divided into four SAC-2 elements, and dynamic simulations were carried out by applying gravitational and air resistance loads within each element. The simulation results were then compared with the experimental results in Figure 10.

The gray line in Figure 10 represents the experimental curve, while the blue and red lines represent the simulation curves before and after 0.5 s, respectively. The pink line shows the motion curve at 0.67 s without air resistance (AR). It can be seen from Figure 10 that considering air damping, a small number of SAC-2 elements can accurately describe the motion state of the flexible free pendulum. Table 10 shows that the maximum root mean square error (RMSE) during the simulation process occurred with the maximum value of 1.5 mm, which is acceptable. By comparing experiments and simulations at 0.67 s, it can be seen that if air resistance is not considered, the resulting RSME is 81.7 mm, which is 54.5 times larger than that of considering air resistance. This demonstrates the necessity of considering air resistance in free pendulum problems.

In this chapter, simulation of the free pendulum with air resistance is conducted, where the height and flexibility of the pendulum are affected by the air resistance. An experiment is performed to validate the simulation results of the free pendulum. The experimental and simulation results indicate that the entire process under air resistance can be described with relatively high accuracy by the space-time ANCF elements, and it is recommended to set the drag coefficient to 1.5 for optimal results.

**Table 10.** The root mean square error in different time.

| Time (s) | 0.01 | 0.19 | 0.30 | 0.42 | 0.53 | 0.67 | 0.79 |
|---|---|---|---|---|---|---|---|
| RMSE (mm) | 0.4145 | 0.3821 | 0.5058 | 0.6869 | 0.6761 | 1.5226 | 0.7383 |

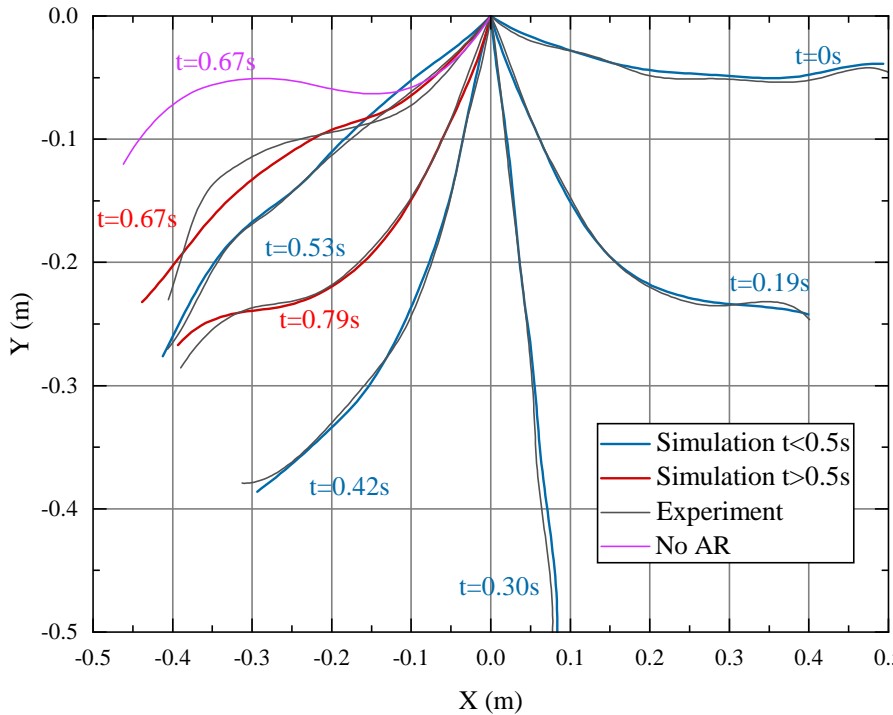

**Figure 10.** The comparison between the fitting curve of experiment and the simulation curve in different times.

## 6. Conclusions

The space-time element based on the absolute nodal coordinate formulation is established in this paper by separately constructing the shape function of time and space directions and taking their product. The resulting element has good conservation properties, as demonstrated by utilizing the Hamiltonian varying action. Two non-conservative forces, bending moment and air resistance, are introduced in this element. To address constraints, a constraint conversion method is proposed by introducing the concepts of replaceable constraint and supplementary constraint to solve the special constraints at the import and export time. Two solvers, utilizing the Newton–Raphson and Broyden rank 1 methods, respectively, are proposed in this paper. In static circle-bending examples, the SAC-3 element is demonstrated to be more accurate than the SAC-2 element while utilizing fewer elements. The stress wave distribution and energy convergence are verified through simulating double-ended velocity impact examples using different discretization elements. Nine combinations are used to analyze the four possible factors of space-time finite element computation efficiency, and their influence on both the total calculation time and part of the calculation time is presented. In non-conservative forces, the accuracy of the SAC element with air resistance is verified through simulation and experiment of a free pendulum example. It is demonstrated that air resistance must be considered in this example, as evidenced by the maximum root mean square error of 1.5 mm with a drag coefficient of 1.5, which represents a good fit between the experiment and simulation.

**Author Contributions:** Conceptualization, D.C.; methodology, D.C.; software, D.C.; validation, D.C.; writing—original draft preparation, D.C.; writing—review and editing, D.C. and K.L.; visualization, D.C.; supervision, N.L.; project administration, N.L.; funding acquisition, N.L. and P.L. All authors have read and agreed to the published version of the manuscript.

**Funding:** The work is funded by Independent Research and Development project of State Key Laboratory of Green Building in Western China [No. LSZZ202209].

**Data Availability Statement:** The data presented in this study are available on request from the corresponding author. The data are not publicly available due to privacy.

**Conflicts of Interest:** The authors declare no conflict of interest.

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
