# Peer review of "A Space-Time Absolute Nodal Coordinate Formulation Cable Element and the Study of Its Accuracy and Efficiency"

_machines, doi:10.3390/machines11040433_

Round 1

Reviewer 1 Report

    Dear Editor 

    Machines-2297154

     Title: A space-time absolute nodal coordinate formulation cable element and the study of its accuracy and efficiency

     Authors: Dekun Chen, Kun Li, Nianli Lu, Peng Lan

    In this paper, authors proposed a space-time absolute nodal coordinate cable element establishing method based on Lagrange family shape function. Two different space-time absolute nodal coordinate formulation cable (SAC) elements with different spatial shape function are proposed by this method. They also presented the external forces such as the bending moment and the air resistance formula. The Lagrange multiplier method and the concepts of replacement constraint and supplementary constraint are introduced, for the solution of dynamics of constrained mechanical systems. The constraint conversion is proposed in the meantime. The solver is established through Hamilton's law of varying action. The space-time finite element method can be used to solve dynamic problems by Newton algorithm and quasi-Newton algorithm. The accuracy and efficiency of the solution are verified with three simulations and one experiment. A circle-bending static simulation and a double-ended velocity impact dynamic simulation shows the accuracy of the two elements. The correlation between Statics and dynamics has been studied for different discretization methods and different solvers' calculation accuracy and efficiency. Different modeling method, time steps, order and the use of quasi-Newton method will influence the efficiency of solution. Comparison with experiment the free-pendulum simulation has ability to simulate the dynamic problems with air resistance.

    Comments 

    The paper makes a significant contribution towards the literature. It is well integrated, interesting and up to date with the existing body of literature. The originality and quality is good. I believe a large audience will gain benefit from it.

    There are some typo mistakes in the paper, like  each mathematical equation should end with "." instead of ",", etc. So authors should read the paper carefully.

    Authors should propose some future work in this directrion.

    Recommendation: Accept after minor revision. 

Reviewer 2 Report

A space-time absolute nodal coordinate cable element establishing method based on Lagrange family shape function is proposed in this work. The nonlinear equations derived from the space-time finite element method are solved by Newton algorithm and quasi-Newton algorithm. The topic is interesting. Here are the detailed comments.

1). In section 2. Establishment of SAC element, please discuss whether the new space-time discretization scheme is symplectic preserving.

2). Newton method and its variant is a powerful tool to solve the nonlinear equation from nonlinear PDE. Please make a brief overview about the recent developments and applications of Newton method and its variant in different scientific communities. Here are some examples, which may useful. [1] Steady-State Analysis of Electrical Networks in Pandapower Software: Computational Performances of Newton–Raphson, Newton–Raphson with Iwamoto Multiplier, and Gauss–Seidel Methods, 2022 [2] An assessment of coupling algorithms in HTR simulator TINTE, 2018 [3] Parallel inexact Newton–Krylov and quasi-Newton solvers for nonlinear elasticity, 2022

3). In section 4, please supplement the convergence history of Newton iteration, and evaluate the convergence rate of Newton iteration. I expect that a super-linear convergence rate could be achieved.

 4).  In section 4, please supplement the order of accuracy in time and in space by refining the size of time step and space step.

Reviewer 3 Report

Please see attached PDF

Round 2

Reviewer 2 Report

No further comment

Author Response

Dear editor and reviewer, thank you very much for taking your time to review this manuscript and we appreciate your approval to our manuscript. Your suggestions really help us a lot and make us clearer about what we can do in the future work.

Reviewer 3 Report

Please see attached PDF
